# Microbe-Derived Antioxidants Alleviate Liver and Adipose Tissue Lipid Disorders and Metabolic Inflammation Induced by High Fat Diet in Mice

**DOI:** 10.3390/ijms24043269

**Published:** 2023-02-07

**Authors:** Qingying Gao, Zhen Luo, Sheng Ma, Chengbing Yu, Cheng Shen, Weina Xu, Jing Zhang, Hongcai Zhang, Jianxiong Xu

**Affiliations:** Shanghai Key Laboratory of Veterinary Biotechnology/Shanghai Collaborative Innovation Center of Agri-Seeds, School of Agriculture and Biology, Shanghai Jiao Tong University, Shanghai 200241, China

**Keywords:** Microbe-derived antioxidants, liver, epididymal adipose tissues, oxidative stress, lipid metabolism disorders, meta-inflammation

## Abstract

Obesity induces lipodystrophy and metabolic inflammation. Microbe-derived antioxidants (MA) are novel small-molecule nutrients obtained from microbial fermentation, and have anti-oxidation, lipid-lowering and anti-inflammatory effects. Whether MA can regulate obesity-induced lipodystrophy and metabolic inflammation has not yet been investigated. The aim of this study was to investigate the effects of MA on oxidative stress, lipid disorders, and metabolic inflammation in liver and epididymal adipose tissues (EAT) of mice fed with a high-fat diet (HFD). Results showed that MA was able to reverse the HFD-induced increase in body weight, body fat rate and Lee’s index in mice; reduce the fat content in serum, liver and EAT; and regulate the INS, LEP and resistin adipokines as well as free fatty acids to their normal levels. MA also reduced de novo synthesis of fat in the liver and EAT and promoted gene expression for lipolysis, fatty acid transport and β-oxidation. MA decreased TNF-α and MCP1 content in serum, elevated SOD activity in liver and EAT, induced macrophage polarization toward the M2 type, inhibited the NLRP3 pathway, increased gene expression of the anti-inflammatory factors IL-4 and IL-13 and suppressed gene expression of the pro-inflammatory factors IL-6, TNF-α and MCP1, thereby attenuating oxidative stress and inflammation induced by HFD. In conclusion, MA can effectively reduce HFD-induced weight gain and alleviate obesity-induced oxidative stress, lipid disorders and metabolic inflammation in the liver and EAT, indicating that MA shows great promise as a functional food.

## 1. Introduction

A high fat diet (HFD) easily induces excessive deposition or abnormal distribution of lipids in the body, causing obesity. In the last two decades, the global obesity rate has increased by about 1.5 times in adults and 2 times in children, with nearly two thirds of adults and one third of children in the European region being overweight or obese [1]. In China, more than 50% of adults (≥18 years old) are already overweight or obese, with 34.3% of adult residents being overweight and 16.4% being obese [2]. Obesity, which increases the risk of developing metabolic diseases such as hypertension (HBP), insulin resistance (IR) and nonalcoholic fatty liver disease (NAFLD) [3,4,5], has become a global public health problem that endangers human health.

Obesity is closely related to increased oxidative stress, lipid disorders and chronic inflammatory response [6,7,8], and the liver, adipocytes, immune cells and other systems are involved in the development and progression of obesity [9]. Metabolic diseases are associated with lipid and energy metabolism of various tissues, and liver and adipose tissue have important roles in maintaining body energy homeostasis [10,11]. Obesity accompanies liver steatosis and, in severe cases, leads to impaired liver function, causing steatohepatitis, which further triggers cirrhosis and even liver cancer [12]. Accumulation of fat in adipose tissue is thought to be associated with insulin resistance and increased risk of metabolic diseases, and meta-inflammation exhibits a high degree of complexity in adipose tissue [13]. Therefore, modulating lipid disorders in the liver and adipose tissue can alleviate metabolic diseases resulting from obesity [14]. Although the treatment of obesity through a combination of diet and drugs has been reported, different approaches still remain ambiguous due to the inability to determine their safety and efficacy. Thus, the search for pharmacological alternatives or nutritional supplements to alleviate obesity or prevent obesity-induced metabolic diseases is urgently needed [15].

Bioactive substances and plant extracts have been reported to have protective and beneficial effects in weight loss and diet-induced metabolic disorders [16,17]. Postbiotics have immunomodulatory and gut barrier-protective effects, and can alleviate insulin resistance and improve glucose tolerance in obese mice [18,19]. The plant extract berberine can alleviate HFD-induced disturbance of the intestinal flora in rats, alleviate the body’s inflammatory response and reduce blood glucose [20]. Our previous studies have shown that novel microbe-derived antioxidants (MA) obtained by probiotic fermentation contain more than 400 small molecule components [21] and have strong free radical scavenging ability [22], as well as anti-stress [22], lipid-lowering and anti-inflammatory effects [23]. They can regulate apoptosis and autophagy [23], maintain hepatocyte function [23,24] and improve the fertilization ability of men with asthenozoospermia [25]. Recently, we showed that MA may regulate Nrf2-ROS-NLRP3-IL-1β signaling pathways to protect cells from LPS-induced oxidative stress and inflammatory responses [26]. However, the effects of MA on HFD-induced obesity and the accompanying responses are unknown. The aim of this study was to investigate the effects of MA on HFD-induced oxidative stress, lipid disorders and inflammatory responses in the liver and adipose tissues of mice.

## 2. Results

### 2.1. Body Weight, Energy Intake, Body Fat Rate and Organ Index Changes in Mice

The body weight of the mice fed with HFD accelerated from the first week onward, and the difference was significant compared with the CON mice (*p* < 0.01). The body weight of the HFD mice at the end of the experiment was 17.67% higher than that of the CON mice, and the difference was significant (*p* < 0.001). Compared with the HFD group, the body weight of the HMA group decreased significantly after week 7 (*p* < 0.05), and the body weight of the HMA group was 9.05% lower than that of the HFD group at the end of the trial, with a significant difference (*p* < 0.001) (Figure 1b). During the experiment, the food intake of the HED group significantly decreased (*p* < 0.001), and the energy intake of the HFD group was 13.01% higher than that of the CON group (*p* < 0.001). The food intake and energy intake of the HMA group were similar to the HFD group, with no significant difference between the two groups (*p* > 0.05) (Figure 1c,j). Compared with the CON group, the abdominal girth (*p* < 0.01), body fat rate (*p* < 0.01), inguinal WAT index (*p* < 0.01), epididymal WAT index (*p* < 0.001) and perirenal WAT index (*p* < 0.001) of mice in the HFD group were significantly increased, while the abdominal girth of mice in the HMA group was significantly decreased compared with the HFD group (*p* < 0.05). Body length was significantly increased (*p* < 0.05), while Lee’s index (*p* < 0.05) and liver index (*p* < 0.001) were significantly decreased (Figure 1d–g). These results indicated that MA was able to alleviate HFD-induced body weight and reduce liver weight, body fat percentage and Lee’s index in mice.

### 2.2. Lipid Metabolism and Inflammatory Response in Serum of Mice

Mice in the HFD group showed a significant increase in serum levels of TG (*p* < 0.001), TC (*p* < 0.001) and LDL-C (*p* < 0.01), and a significant decrease in HDL-C content (*p* < 0.05), which was reversed in the HMA mice (Figure 2a−d). The contents of INS, resistin and LEP in the serum of the HFD mice were significantly increased (*p* < 0.001), and ADP was significantly decreased (*p* < 0.001). Compared with the HFD group, mice in the HMA group had significantly lower serum levels of resistin (*p* < 0.001) and LEP (*p* < 0.01), and higher levels of ADP (*p* < 0.01) (Figure 2e−h). The activities of ALT and AST were significantly increased in the serum of the HFD mice (*p* < 0.001), while the activities of ALT and AST were significantly decreased in the serum of the HMA mice (*p* < 0.001) (Figure 2i,j). The serum MCP1 and TNF-α contents in the HFD mice were significantly increased compared with those in the CON mice (*p* < 0.001), while those in the HMA mice were decreased by 9.12% (MCP1) and 13.12% (TNF-α), respectively, compared with those in the HFD group (*p* < 0.05) (Figure 2k,l). These results indicated that MA alleviated lipid disorders and liver dysfunction and reduced inflammatory responses induced by the HFD.

### 2.3. Oxidative Stress in Liver and EAT of Mice

The supplementation of MA in the basal diet significantly increased the activity of SOD and content of GSH in the livers of mice (*p* < 0.05), and the content of MDA in the liver was significantly increased (*p* < 0.001). The activities of CAT (*p* < 0.001) and SOD (*p* < 0.01), as well as the contents of GSH (*p* < 0.001), were significantly decreased. Compared with the HFD group, the activity of SOD and contents of GSH were significantly increased by MA (*p* < 0.05) (Figure 3A−D). Similarly, MDA content in the EAT of the CMA group was significantly decreased (*p* < 0.05). MDA content in the HFD group was significantly increased (*p* < 0.001), and the activity of CAT and SOD (*p* < 0.05), as well as the contents of GSH (*p* < 0.01), were significantly decreased. After MA treatment, MDA content in the HMA group was significantly decreased (*p* < 0.001), and CAT (*p* < 0.01) and GSH (*p* < 0.05) activities were significantly increased (Figure 3E−H). These results suggest that MA could improve the antioxidant capacity of the liver and EAT in mice, and could also alleviate the oxidative damage induced by HFD.

### 2.4. Lipid Metabolism in Liver and EAT of Mice

HE staining and oil red staining of the liver sections showed that there were obvious fat vacuoles and red fat droplets in the livers of the HFD group. MA improved the structural damage to the liver that was induced by HFD (Figure 4A). The contents of TG (*p* < 0.001), TC (*p* < 0.01) and LDL-C (*p* < 0.001) in the livers of the HFD group were significantly increased. The expressions of CD36 (*p* < 0.05), CPT1b (*p* < 0.01) and ATGL (*p* < 0.001) were decreased, and the expression of FAS (*p* < 0.01) was increased. The expression of the FABP1 (*p* < 0.01) and HSL (*p* < 0.01) genes were significantly increased in the CMA group. Compared with the HFD group, the contents of TG (*p* < 0.001), TC (*p* < 0.05) and LDL-C (*p* < 0.001) were significantly decreased in the HMA group, and the expression of the CD36 (*p* < 0.01), CPT1b (*p* < 0.05) and ATGL (*p* < 0.001) genes were increased. The expressions of the FAS and ACCα genes were decreased (p < 0.05) (Figure 4C–E). In EAT, compared with the CON group, the contents of TG (*p* < 0.001), TC (*p* < 0.01) and LDL-C (*p* < 0.05) in the HFD group were significantly increased. The expressions of the FABP1 (*p* < 0.001), CPT1b (*p* < 0.05) and ATGL (*p* < 0.05) genes were significantly decreased, and the gene expressions of FAS were significantly increased (*p* < 0.01). ATGL gene expression was significantly increased in the CMA group (*p* < 0.001). Compared with the HFD group, the HMA mice showed significantly decreased TG and TC contents, increased FABP1 (*p* < 0.05) and ATGL (*p* < 0.001) gene expression and decreased ACCα gene expression (*p* < 0.01) (Figure 4C,G−I). These results suggest that MA could promote fatty acid transport, β-oxidation and esterification in the liver and EAT, and could inhibit fatty acid synthesis.

### 2.5. Inflammatory Reaction in Liver and EAT of Mice

The expressions of NLRP3, Caspase1, IL-18 and IL-1β in the liver and EAT of the HFD mice were significantly increased (*p* < 0.001). Confocal laser microscopy showed that the fluorescence signal of the M2-type macrophage marker CD163 was decreased. Because the M1 and M2 phenotypes are continuous, the (CD86 + CD68) double label was used to represent M1-type macrophages, and (CD163 + CD68) double label was used to represent M2-type macrophages. CD68 is the most widely used marker of macrophages, while CD86 and CD163 are markers of M1 and M2 polarization for macrophages, respectively. The average fluorescence intensity of (CD163 + CD68)/(CD86 + CD68) was significantly decreased (*p* < 0.01), and the gene expressions of IL-6, TNF-α, MCP1, IL-4, IL-10 (*p* < 0.001) and IL-13 (*p* < 0.05) were significantly increased. The expression of the PPARγ gene was significantly decreased in the liver (*p* < 0.001). The expressions of the MCP1 (*p* < 0.05), IL-10 and IL-13 (*p* < 0.01) genes were significantly increased, and the expressions of the PPARα (*p* < 0.01) and PPARγ (*p* < 0.001) genes in EAT were significantly decreased. Compared with the HFD group, HMA mice showed significantly reduced expression of the NLRP3, IL-18, IL-1β, TNF-α, MCP1, Caspase1 and IL-6 genes, and increased expression of the PPARγ (*p* < 0.001), IL-4 (*p* < 0.05) and IL-13 (*p* < 0.01) genes in the liver (Figure 5a−c). Meanwhile, compared with the HFD group, the HMA mice showed significantly decreased expressions of the NLRP3, Caspase1 (*p* < 0.05), IL-1β (*p* < 0.01), IL-6 (*p* < 0.05), TNF-α (*p* < 0.001) and MCP1 (*p* < 0.05) genes, and significantly increased expression of the ASC gene (*p* < 0.01) in EAT (Figure 5d−f). Confocal laser microscopy showed that the fluorescence labeling of CD163 in liver tissue and EAT M2-type macrophage markers was increased, and the average fluorescence intensity of (CD163 + CD68)/(CD86 + CD68) was significantly increased (*p* < 0.01) (Figure 5g,h).

## 3. Materials and Methods

### 3.1. Animal Grouping and Feeding Management

A total of 40 6-week-old C57BL/6J male mice (weighing 18–22 g) were purchased from Shanghai JieSiJie Laboratory Animal Co., Ltd. (Shanghai, China). After pre-feeding for one week, the mice were randomly divided into 4 groups: control group (CON), control +MA group (CMA), high-fat group (HFD) and high-fat +MA group (HMA). The CON and CMA groups were fed a conventional diet (10% fat, 1.61 × 10^7^ J/kg, D12492J), while the HFD and HMA groups were fed a high-fat diet (60% fat, 2.19 × 10^7^ J/kg, D12492). Both the conventional diet and the high-fat diet were purchased from Jiangsu Xietong Pharmaceutical Bio-engineering Sybiologic Co., Ltd. The CMA and HMA groups received a daily gavage of 0.03 mL MA/10 g body weight, while the CON and CMA groups received the same amount of saline instead of MA. MA was provided by Jianghan Biotechnology (Shanghai) Co., Ltd. (Trade name KB-120, batch number: 22129142), and the main ingredients of MA were described by Luo et al. (2022) [21]. The mice were kept in a plastic cage of 25 cm × 40 cm with a stainless steel cage lid under a standard 12 h light/12 h dark cycle. The temperature was controlled at 20–25 °C, and the relative humidity was controlled at 50–60%. The mice were allowed to drink and eat freely. Weight and feed intake were measured every 2 days for 12 weeks. The food intake energy levels of the different groups of mice were calculated according to the calories in the diet. The animal study protocol was approved by the Shanghai Jiao Tong University Institutional Animal Care and Use Committee. The ethical approval code for the manuscript is 202201188.

### 3.2. Sample Collection and Preparation

At the end of the experiment, in the morning when the stomachs of the mice were empty, the eyeballs of the mice were removed for blood collection. The blood was collected in a 1.5 mL centrifuge tube and placed at room temperature for 30 min. The blood was transferred to a 4 °C refrigerator for 4 h, centrifuged at 3500× *g* for 20 min at 4 °C, and the supernatant was transferred to a 200 μL centrifuge tube and stored in a −20 °C freezer (Qingdao Haier Special Electric Appliance Co., Ltd., Qingdao, China) for later use. After blood collection, the mice were decapitated, and the abdominal girth and body length of the mice were measured with a soft ruler. The abdominal girth was measured with a soft ruler that passed through the back of the mouse and then around the abdomen, and the body length was determined as the length from the nose to the anus. After dissection, the organs of the mice were removed, and tissue samples, including liver, kidney, spleen, lung, heart, inguinal adipose tissue (IAT), epididymal adipose tissue (EAT) and perirenal adipose tissue (PAT) were collected. All three types of adipose tissue were white adipose tissue (WAT). Connective tissue was removed from each organ, and the adipose tissues were rinsed with pre-cooled saline, then dried with filter paper, weighed, placed into a frozen tube, frozen in liquid nitrogen and stored in a −80 °C freezer. The livers and EAT were weighed and homogenized in normal saline (1:9, *w/v*). After centrifugation at 5000× *g* for 10 min at 4 °C, the supernatant was collected and stored in a −80 °C freezer (Qingdao Haier Special Electric Appliance Co., Ltd., Qingdao China). The protein concentration was determined according to the instructions of the BCA protein assay kit (P0010, Beyotime Biotech, Shanghai, China). The index of visceral tissue and WAT was calculated by dividing the weights of the different visceral and adipose tissues by the body weight and multiplying by 100%. The Lee’s index was calculated using the following formula: weight (g) ^ (1/3) × 10/body length (cm). The Lee’s index comprehensively reflects the proportional relationship between body weight and body length, and can better reflect the degree of obesity of rats/mice. The body fat rate was calculated using the following formula: (EAT weight (g) + PAT weight (g))/body weight (g) × 100%. 

### 3.3. H&E Staining and Oil Red Staining Were Performed

Fresh livers were fixed in 4% paraformaldehyde, dehydrated and embedded in paraffin to prepare sections. The livers were stained with hematoxylin and eosin (H&E) to determine the histopathological morphology. The liver samples were also embedded in the frozen liver section for oil red staining to visualize lipid droplets.

### 3.4. Determination of TG, TC, HDL-C, LDL-C and Transaminase Activity

The levels of triglycerides (TG), cholesterol (TC), high-density lipoprotein cholesterol (HDL-C) and low-density lipoprotein cholesterol (LDL-C) in the serum, liver and EAT samples, as well as the activities of aspartate aminotransferase (AST) and alanine aminotransferase (ALT) in serum, were determined using their respective kits per manufacturer’s instructions (Nanjing Jiancheng Bioengineering Institute, Nanjing, China), as described previously [23].

### 3.5. Determination of Oxidative Stress Index

The activities of superoxide dismutase (SOD), malondialdehyde (MDA), glutathione (GSH) and catalase (CAT) in the liver and EAT were determined using assay kits from Nanjing Jiancheng Bioengineering Institute (Nanjing, China) according to the manufacturers’ instructions, as described previously [27].

### 3.6. An ELISA Assay Was Also Utilized

The contents of monocyte chemotactic protein 1 (MCP-1), tumor necrosis factor (TNF-α), insulin (INS), leptin (LEP), resistin and adiponectin (ADP) in serum and cytokines IL-4, IL-6, IL-10, IL-13, TNF-α and MCP1 in the liver and EAT were determined using the enzyme-linked immunosorbent assay (ELISA) kits obtained from the Shanghai Enzyme-linked Biotechnology Co., Ltd. (Shanghai, China). The liver and EAT samples were homogenized in PBS and centrifugated at 12,000× *g* for 15 min. Microplates were coated with antibodies for each factor and then detected with horseradish peroxidase-labeled substrates after 10 min of incubation at 37 °C. The absorbance values were read at 450 nm in a spectrophotometer.

### 3.7. Immunofluorescence

Paraffin-embedded liver and EAT were prepared as 5 μm sections on glass slides. The slides were incubated in 0.3% H_2_O_2_ in methanol to inhibit endogenous peroxidase activity, followed by blocking with 1% BSA and 1% normal goat serum in PBS for 10 min. The slides were then incubated with rabbit anti-CD68 monoclonal antibody (1:200, 97778, CST, Boston, MA, USA) overnight at 4 °C. The secondary antibody (K5007, DAKO, Copenhagen, Denmark) was incubated at room temperature for 1 h. TSA CY3 (SH0002, Runnerbio, Shanghai, China) was incubated at room temperature for 30 min. Subsequently, the rabbit anti-CD163 monoclonal antibodies (1:200, ab182422, Abcam, Cambridge, UK) + TSA 488 (SH0001, Runnerbio, Shanghai, China) were sequentially added. In addition, the rabbit anti-CD86 monoclonal antibodies (1:150, 19589, CST, Boston, MA, USA) +TSA 488 (SH0001, Runnerbio, Shanghai, China) were sequentially added. Finally, sections were incubated with DAPI (1:500, Basel, Roche) at room temperature for 20 min. Three fields/sample from three samples were randomly selected for each group. The mean gray value was analyzed using Image J (National Institute of Mental Health, Bethesda, MD, USA) (Mean gray value = Integrated Density/Area).

### 3.8. Reverese Transcription–Quantitative Real-Time Polymerase Chain Reaction (RT-qPCR)

The total RNA from the liver and EAT was isolated using the Tissue RNA Purification Kit PLUS (EZBioscience, Roseville, CA, USA), following the manufacturer’s instructions. The amount and quality of the RNA were determined using a spectrophotometer (Thermo Fisher Scientific, Waltham, MA, USA). The Color Reverse Transcription Kit (EZBioscience, Roseville, CA, USA) was used to convert 1 μg of RNA into cDNA. Gene expression was quantified by RT-qPCR reaction using the Light Cycler96 system (Roche). RT-qPCR was performed according to the instructions provided by the 2 × Color SYBR Green qPCR Mix kit (EZBioscience, Roseville, CA, USA). The total volume of the reaction was 20 μL, including 10 μL SYBR Green mix, 2 μL cDNA, 7.2 μL H_2_O and 0.4 μL each of forward and reverse primers. The amplification conditions included denaturation at 95 °C for 5 min, followed by 40 cycles of 95 °C for 10 s and 60 °C for 30 s. Finally, a melting curve analysis was performed. The primers are listed in Appendix A. β-actin was used as a housekeeping gene to normalize target gene transcript levels. The values were expressed using the formula 2^−(ΔΔCt)^, where ΔΔCt = (Ct Target-Ct β-actin) treatment—(Ct Target-Ct β-actin) model.

### 3.9. Data Analysis

Data were expressed as mean ± SEM. The statistical software SPSS 20.0 (SPSS Inc., Chicago, IL, USA) was used for data analysis. Multiple comparisons were performed using one way analysis of variance (ANOVA), followed by LSD as a post hoc test. *p* < 0.05 was considered statistically significant, and *p* < 0.001 was considered highly statistically significant.

## 4. Discussion

A long-term high-fat diet can easily cause imbalances in energy intake and consumption, resulting in lipid metabolism disorders and obesity and thus increasing the incidence of NAFLD, HBP, IR and other metabolic syndromes [28]. In this study, it was found that dietary supplementation of MA reduced body weight gain and liver fat deposition induced by HFD in mice, as well as decreased body fat percentage and Lee’s index. However, there was no significant difference in energy intake, suggesting that the effect of MA may be related to the regulation of fat metabolism. The level of blood lipids is a direct reflection of the body’s lipid metabolism [29]. We further detected the level of serum lipids in mice and found that MA reduced the contents of serum TG, TC and LDL-C induced by HFD in mice. These results suggested that MA inhibited HFD-induced liver fat content, thereby reducing body weight, inhibiting lipid accumulation and alleviating metabolic disorders.

Fat is the main source of energy for animals, and the fatty acids that make up fat mainly come from food intake and de novo synthesis. No significant increase in energy intake was observed in this experiment, but it was found that the addition of MA decreased the gene expression levels of fatty acid synthase (FAS) and acetyl CoA carboxylase α (ACCα) in the liver and EAT of mice. Liver and adipose tissue are major sites responsible for lipid synthesis, transport and oxidation decomposition, and FAS and ACCα are mainly involved in de novo fatty acid synthesis. Previous studies have shown that MA could reduce fat synthesis in the liver and EAT [30,31,32]. The decomposition of fat can also affect the storage of fat in liver and adipose tissue. Lipid catabolism requires the induction of lipid-metabolizing enzymes, such as adipose lipide lipase (ATGL), hormone-sensitive lipase (HSL) and monoacylglycerol lipase (MAGL). Fatty acids produced by lipolysis bind to albumin to form HDL-C, which is transported via blood circulation to energy-requiring tissues, where fatty acids are taken up by various transporters [33]. Fatty acid translocase (CD36) and fatty acid-binding protein 1 (FABP1) are essential for fatty acid uptake and transportation. In this study, the expression of the CD36 gene in the livers of the HFD mice was significantly reduced. After feeding them a high-fat diet and supplementing with MA, the expression of the CD36 gene was significantly increased to a level similar to that of the CON group. This indicated a restriction in fat mass gain when fed HFD. Some fatty acids can be used for the synthesis of other lipids or for oxidative decomposition in tissues, among which carnitine palmitoyl transterase1 (CPT1) is an important rate-limiting enzyme involved in fatty acid uptake and fatty acid β-oxidation in mitochondria. In this study, the addition of MA significantly increased the expression of the CD36, CPT1b and ATGL genes in mouse livers, and that of the FABP1 and ATGL genes in EAT. Notably, it was also found in this study that the expression of the ATGL gene in the liver and EAT of mice fed MA was significantly increased. As the most important lipolysis enzyme, a reduction in ATGL expression will lead to massive accumulation of TG in adipose cells and other tissues, resulting in obesity and other metabolic complications [34,35]. The above analysis showed that MA could inhibit fat synthesis in liver and adipose tissue, promote fat decomposition and transport and reduce fat deposition, thus reducing weight gain. It is speculated that MA may inhibit the inflammation induced by a high-fat diet, thus regulating disorders of fat metabolism in the body.

In recent years, more and more studies have found that adipokines are involved in the regulation of fat metabolism, and the disorder of adipokine synthesis and secretion is closely related to inflammation and abnormal fat deposition. Adipokines can affect insulin sensitivity and lipid metabolism in the liver, and can indirectly affect inflammation and oxidative stress signaling pathways that are closely related to NASH. Meanwhile, adipokines can also directly activate or inhibit related signaling pathways, affecting inflammation and other processes, and thus affecting the disease course of NAFLD [36,37]. In this study, it was found that dietary supplementation with MA decreased the serum levels of proinflammatory adipokine LEP, resistin, TNF-α and MCP1, and increased the levels of anti-inflammatory adipokine ADP in mice fed high-fat diets, indicating that MA reduces inflammation.

As a self-protective mechanism, inflammation itself plays an important role in lipid metabolism and intracellular lipid accumulation. Our previous study found that MA was able to alleviate DSS-induced intestinal inflammation through NOD-like receptor thermal protein domain associated protein 3 (NLRP3) inflammasome [38]. Inflammatory cytokines mediated by NLRP3 can be activated in various cells in metabolic tissues and lead to metabolic disorders. Obesity is one of the most common metabolic diseases [39,40]. Clinical studies have shown that the NLRP3 pathway is significantly activated and the levels of Caspase1 and IL-1β are significantly increased in obese and type 2 diabetes patients compared to healthy subjects [41]. Caloric restriction- and weight loss-induced insulin sensitivity was associated with decreased NLRP3 and IL-1β expression in subcutaneous adipose tissue, further suggesting that NLRP3 is associated with diabetes and obesity [42]. These results indicate that lipid metabolism disorders lead to inflammation, and that MA can reduce lipid accumulation and alleviate inflammation. In order to further confirm the hypothesis, the levels of relevant inflammatory factors in the liver and EAT were measured in this study, and it was found that dietary addition of MA could significantly inhibit the NLRP3 pathway and the gene expressions of pro-inflammatory factors IL-6, TNF-α and MCP1 in the liver and EAT. It could also significantly increase the gene expressions of anti-inflammatory factors IL-4 and IL-13. Sun et al. (2022) found that Ramulus Mori (Sangzhi) alkaloids achieved the amelioration adipose tissue inflammation by reducing macrophage infiltration in adipose tissue, which, in turn, significantly inhibited the expression of pro-inflammatory factors TNF-α and MCP1 and up-regulated the gene transcription of anti-inflammatory factors IL-4, IL-10 and IL-13 [43]. We found that MA increased the expression of M2-type macrophage marker CD163 in liver tissue and EAT, and the average fluorescence intensity of (CD163 + CD68)/(CD86 + CD68) was significantly increased. CD68 is the most widely used marker of macrophages, while CD86 and CD163 are markers of M1 and M2 polarization of macrophages, respectively. Macrophages are major players in the pathogenesis of many chronic inflammatory and immune diseases and can be activated into different inflammatory states under stimulation, including M1 (classical activation) and M2 (selective activation) [44,45,46]. In the process of inflammation, M1-type macrophages release IL-6, IL-1β, ROS, nitric oxide, etc., to enhance phagocytosis and promote an inflammatory response. In the stage of inflammation regression, M2-type macrophages play a dominant role and activate IL-4, IL-10, IL-13 and other cytokines, showing a high anti-inflammatory and antioxidant capacity. They have a restorative effect on tissues [45,47]. Because the M1 and M2 phenotypes are continuous, the (CD86 + CD68) double label is used to represent M1-type macrophages, and the (CD163 + CD68) double label is used to represent M2-type macrophages [48]. The proliferator-activated receptors (PPARs) may be involved in the initiation of anti-inflammatory signals of M2 macrophages. It has been reported that PPARγ activation can promote the polarization of macrophages and up-regulate the expression of anti-inflammatory factors such as CD206, CD163 and IL-10 [49,50]. Odegaard et al. reported that the number of M2 macrophages in bone marrow PPARγ knockout mice decreased, and the risk of obesity and insulin resistance induced by a high-fat diet increased. Their results showed that PPARγ is required for simulation of alternatively activated macrophages [51]. In this study, the expression of the PPARγ gene in liver and epididymal adipose tissue was significantly reduced in the HFD group, and MA was able to significantly increase the expression of the PPARγ gene in the liver. PPARγ may be involved in the initiation of anti-inflammatory signals of M2 macrophages. The results of this study indicate that MA could inhibit the levels of NLRP3 inflammasomes, reduce HFD-induced liver inflammation, induce M2-type polarization of macrophages and alleviate metabolic inflammation, but the mechanism requires further study.

Oxidative stress is closely related to inflammation. A large number of studies have shown that ROS is needed to activate the NLRP3 inflammasome, and the accumulation of ROS in vivo or cells will cause oxidative damage [52,53]. MA is a novel multi-molecular mixture which is produced from sea buckthorn and rose by microbial fermentation. Sea buckthorn contains a large number of lipophilic antioxidants (mainly tocopherols and carotenoids) and hydrophilic antioxidants (flavonoids, phenolic acids and ascorbic acid). There are many active compounds in Roxthorn rose, including vitamin C, SOD and other antioxidants. It has a higher antioxidant capacity than other common fruits and vegetables [54,55]. Recently, we found that MA may protect cells from LPS-induced oxidative stress and inflammation by regulating the Nrf2-ROS-NLRP3-IL-1β signaling pathway. In this study, it was found that MA reversed MDA and GSH content, as well as CAT and SOD activities, that were induced by HFD. The antioxidant capacity was effectively improved and oxidative stress was alleviated, indicating that MA may protect the liver through the ROS-NLRP3 signaling pathway and may alleviate oxidative stress and inflammation induced by a high-fat diet, thus regulating lipid metabolism disorders.

## 5. Conclusions

The results of this study showed that MA was able to significantly alleviate liver and EAT lipid metabolism disorders induced by a high-fat diet in mice by inhibiting fat synthesis and promoting fat decomposition and transport. It effectively reduced the weight gain induced by the high-fat diet. MA can inhibit oxidative stress in mice, induce macrophages to become M2-type polarized, inhibit the expression of the NLRP3 inflammasome and reduce metabolic inflammation; thus, it has broad prospects in functional foods.

## Figures and Tables

**Figure 1 ijms-24-03269-f001:**
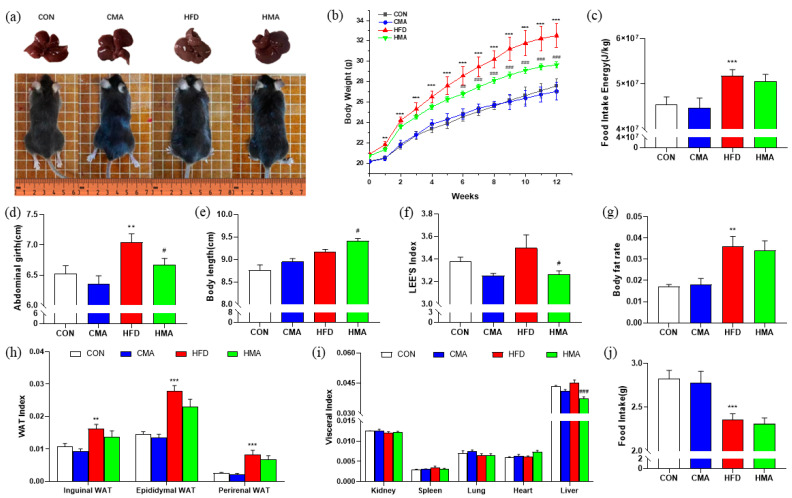
Effects of MA on body weight, energy intake, body fat rate and organ index in mice. (**a**) Mouse appearance and liver. (**b**) Body weight. (**c**) Food intake energy. (**d**) Abdominal girth. (**e**) Body length. (**f**) Lee’s index. (**g**) Body fat rate. (**h**) WAT index. (**i**) Visceral index. (**j**) Food intake. C57BL/6J mice were divided into four groups and fed a basal diet (CON, 10% fat) or high-fat diet (HFD, 60% fat) (CON + 0.03 mL/10 g MA (CMA) and HFD + 0.03 mL/10 g MA (HMA)) for 12 weeks. Data are expressed as mean ± SEM (n = 10). Differences between the CON group and the CMA and HFD groups are indicated by *. *** *p* < 0.001, ** *p* < 0.01. Differences between the HFD group and the HMA group are expressed by ^#^. ^###^ *p* < 0.001, ^##^ *p* < 0.01, ^#^ *p* < 0.05.

**Figure 2 ijms-24-03269-f002:**
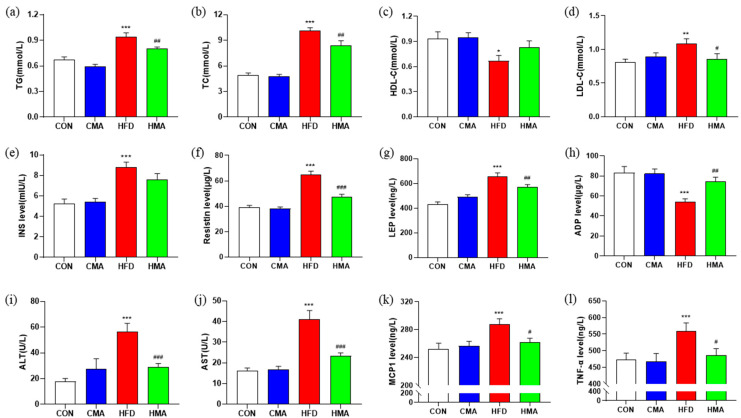
Effects of MA on serum lipid metabolism, liver function and inflammatory response in mice. (**a**) Serum TG. (**b**) Serum TC. (**c**) Serum HDL-C. (**d**) Serum LDL-C. (**e**) Serum INS content. (**f**) Serum resistin content. (**g**) Serum LEP content. (**h**) Serum ADP content. (**i**) Serum ALT activity. (**j**) Serum AST activity. (**k**) Serum MCP1 content. (**l**) Serum TNF-α content. Values are expressed as mean ± SEM (n = 10). Differences between the CON group and the CMA and HFD groups are indicated by *. *** *p* < 0.001, ** *p* < 0.01, * *p* < 0.05. Differences between the HFD and HMA groups are expressed by ^#^. ^###^ *p* < 0.001, ^##^ *p* < 0.01, ^#^ *p* < 0.05.

**Figure 3 ijms-24-03269-f003:**
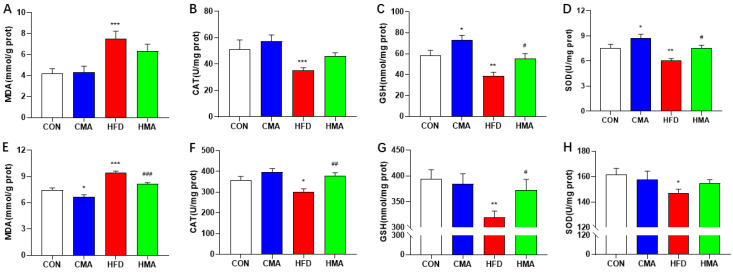
Effects of MA on oxidative stress in liver and EAT of mice. (**A**−**D**) Liver oxidative stress indicators. (**E**−**H**) EAT oxidative stress indicators. Values are expressed as mean ± SEM (n = 10). Differences between the CON group and the CMA and HFD groups are indicated by *. *** *p* < 0.001, ** *p* < 0.01, * *p* < 0.05. Differences between the HFD group and the HMA group are expressed by ^#^. ^###^ *p* < 0.001, ^##^ *p* < 0.01, and ^#^ *p* < 0.05.

**Figure 4 ijms-24-03269-f004:**
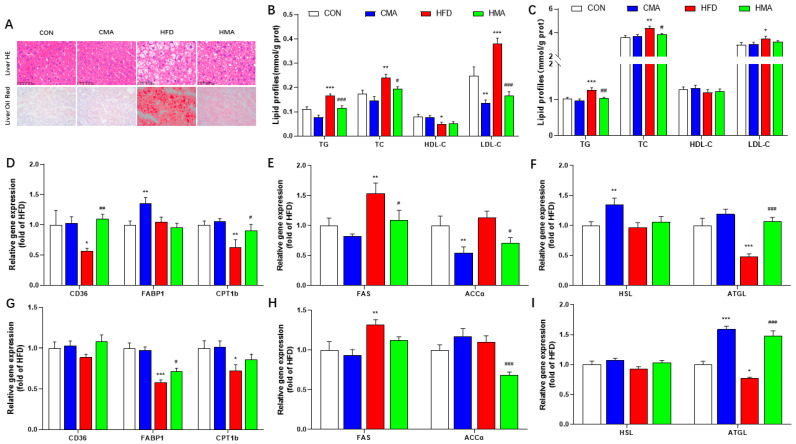
Effect of MA on HFD-induced liver structural damage, and contents of TG, TC, HDL-C, LDL-C, gene expression of lipid transport (CD36 and FABP1), lipases (ATGL and HSL), β-oxidation (CPT1b) and synthesis (FAS and Accα) in the liver and EAT of mice. (**A**) HE staining of liver and EAT (400×). (**B**) Contents of TG, TC, HDL-C and LDL-C in liver tissue. (**C**) Contents of TG, TC, HDL-C and LDL-C in EAT. (**D**) Fatty acid transport and β-oxidation in liver tissue. (**E**) De novo fat synthesis in liver tissue. (**F**) Fat metabolism enzymes in liver tissue. (**G**) Fatty acid transport and β-oxidation in EAT. (**H**) De novo fat synthesis in EAT. (**I**) Fat metabolism enzymes in EAT. The values are expressed as mean ± SEM (n = 10). Differences between the CON group and the CMA and HFD groups are indicated by *. *** *p* < 0.001, ** *p* < 0.01, * *p* < 0.05. Differences between the HFD group and the HMA group are expressed by ^#^. ^###^ *p* < 0.001, ^##^ *p* < 0.01, and ^#^ *p* < 0.05.

**Figure 5 ijms-24-03269-f005:**
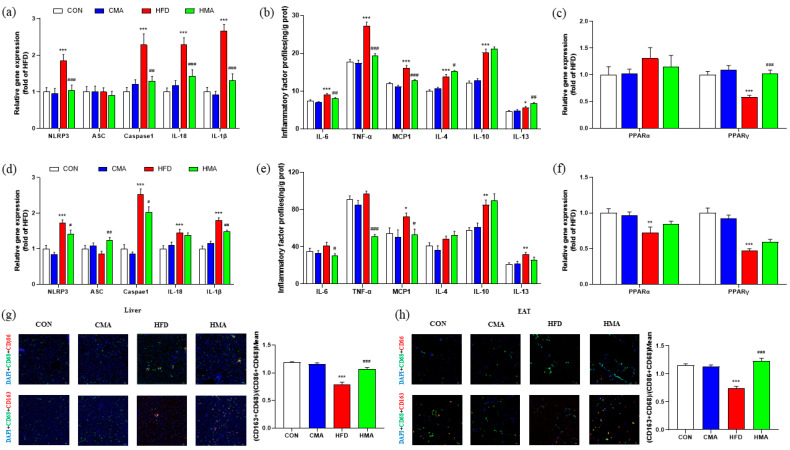
Effect of MA on inflammatory response in liver tissue and EAT of mice. (**a**−**c**) Expression of inflammatory genes and content of inflammatory factors in liver tissue. (**d**−**f**) Expression of inflammatory genes and content of inflammatory factors in EAT. (**g**) Fluorescent labeling of macrophages in liver. (**h**) Fluorescent labeling of EAT macrophages. (**a**−**f**) Values are expressed as mean ± SEM (n = 10); (**g**−**h**) values are expressed as mean ± SEM (n = 3 * 3). Differences between the CON group and the CMA and HFD groups are indicated by *. *** *p* < 0.001, ** *p* < 0.01, * *p* < 0.05. Differences between the HFD group and the HMA group are expressed by ^#^. ^###^ *p* < 0.001, ^##^ *p* < 0.01, and ^#^ *p* < 0.05.

## Data Availability

Not applicable.

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
