# Peer review of "Microbe-Derived Antioxidants Alleviate Liver and Adipose Tissue Lipid Disorders and Metabolic Inflammation Induced by High Fat Diet in Mice"

_ijms, 2023, doi:10.3390/ijms24043269_

Round 1
Reviewer 1 Report
Introduction:
Authors sought to investigate the “effects of MA on HFD- induced oxidative stress, lipid metabolism disorders and inflammatory responses in liver and adipose tissues of mice”. The rationale/motivation for the study is clearly explained.
There is a disconnect between methods and results which makes it difficult to understand reported results without a clear link to methods. Comments are given below.
Methods
*Authors need to check tense of reporting in general – there is a mixture of present tense and instruction statements. Since this was done, past tense is expected for all reporting. I have pointed out some but not all.
Use of word “patients” (2.1)for mice does not seem appropriate. Maybe use animals or just mice.
Responsible animal ethics committee and clearance certificate number is not mentioned.
Indication of caloric value (instead of just fat content) for Con and HFD feed might be useful for understanding caloric intake results (Fig 1)
2.2 -20℃ freezer (not refrigerator)
2.2 After blood collection , mice were decapitated (instead of “After blood collection, mice were removed from the neck and killed.”)
2.2 Use past tense (Connective tissue is removed)
2.2 In the freezer at -80℃ (not refrigerator) * Any temp below 0 is in the freezer. Freezer makes need to be specified.
2.4 Sentence incomplete when it starts with “determine”. Maybe start with “to determine ……. Such such was used as per manufacturer’s instructions” Specify kit/instrumentation which is not done in this case.
Same for 2.5 - past tense and also specify methodology – did you use kits or instrument/meter? Complete sentence for oxidative stress and 2.6 which start with “determine”, “homogenize” – all reporting should be past tense. Review all methods
2.7 am not clear about “implanted”. This “implantation” is given as done for liver only but legend for Figure 3. “Effects of MA on oxidative stress in liver and EAT of mice.” Which is which? Manufacturer of kits/antibodies used for immunofluorescence? Authors gave supplier for ELISA which is good.
Results
Authors need to go through the methods and ensure there is enough details and direction for the author to understand the results. There is a bit of disconnect between methods and reported results – maybe some methods are not fully described. I had to keep going back to methods to figure out results – which should not happen when methods and results are described in full.
- How were the indices calculated? (Lee’s, BW, various fat) these are not mentioned in methods at all other than the fact that they were weighed.
- There is mention in methods that lipid profile and liver enzymes was measured in in serum, liver and EAT according to manufacturer's instructions (2.4 Nanjing Jiancheng Bioengineering Institute, Nanjing, China). But results only report in serum
- There is a description of homogenization of liver and EAT in water – maybe indicate in methods what this was used for.
- For fluorescence there is description of frozen liver but there is no mention of samples saved for freezing in methods and purpose.
- The paraffin-embedded liver and EAT sections processing? Results show Oil red staining for liver which is not in methods. Histology results for EAT? Fig 4
These are just some of the inconsistencies. There are other examples that really need to be addressed to clarify and maintain flow of the manuscripts.
Discussion
Because I had problems relating methods to results reported, I had a problem following the arguments made in the discussion. I therefore did not make specific comments for that section.
Reviewer 2 Report
In this manuscript the authors would like to demonstrate the positive effects of microbe-derived antioxidants (MA) against meta-inflammation induced by high fat diet in mice. In particular, they point to MA being able to reduce lipid metabolism disorders, oxidative stress, and inflammatory response that occur in liver and epididymal adipose tissue (EAT) of obese mice.
The topic is undoubtedly original and interesting, and the experimental results could be significant but, in spite of that, the quality of presentation is very low. Indeed, there are too many English flaws, in particular syntax is often incorrect and make sentences difficult to read. Consequently, I am actually unable to express my opinion on its scientific soundness. Moreover, there are also several orthographic mistakes and/or typos (e.g. patients at line 77 instead of animals, LMA instead of CMA at line 83, and so on).
I think that the manuscript needs a drastic editing revision to make it understandable and allow the reviewing process necessary to suggest its publication or not.
In addition, there are other main aspects that are not clear or must be improved:
- It seems to me that there are important contradictions in lipid metabolism and inflammation data. Factors involved in transport and disposal of lipids favoring their reduction in liver and EAT actually appeared improved by MA, but also esterification that would instead promote lipid accumulation. Also cytokine profile is ambiguous since both up- and down-regulation of pro- and anti-inflammatory factors seem to occur by MA administration;
- “Methods” section is very poor and lacks essential information (e.g. technical approach to assess lipid profile and oxidative stress are not described, immunofluorescence, methods for RNA quantification) and some materials are not specified (e.g. antibody supplier); in this connection are data about ROS reliable? Although fluorescence determination has not been described, I supposed it is aimed at measuring ROS production, but these species are very reactive. Are they still present in animal tissue at the end of treatment? Evaluation of markers of lipid oxidation end-products, such as HNE-adducts, would be more reliable;
- How has been measured the energy intake?
- figure quality is very low and is not able to prove evidence reliability;
- in fig. 3.i DAPI evaluation is reported but it wasn’t mentioned in the text, neither in “Materials” and in “Results”;
- statistical evaluation is not properly reported, e.g. is not always specified among which groups difference significance has been calculated.
- “Supplementary Materials” is missing!
- some references are not easily available (e.g. 16, 21) or do not seem very pertinent (e.g. 6-8, 28, 36), or are too old (e.g. 27, 32)
-
Other minor points:
- please explain what is LEE index in mice;
- fig. 1, do the author mean renal failure index? Fig 1.g, is it reported body fat rate or BMI?
- GSH is not an enzyme!
- Abbreviations must be explained at the first time they are introduced in the text.
In conclusion, I do not think that the manuscript is suitable for publication. I suggest to the authors to resubmit it after an extensive revision.
Round 2
Reviewer 2 Report
In the present version, English has been extensively revised thus allowing manuscript reading and comprehension.
"Methods" also have been more in-depth described, references have been checked and changed when necessary, and other shortcomings have been properly amended, improving the overall quality of the work.
Nevertheless, in my opinion, there are still some major points which must be furtherly clarified before manuscript publication:
- point 3, the authors have not properly answered. Undoubtedly, according to literature, PPARs are involved in inflammatory responses regulation, but the experimental data herein reported showed MA favoured lipid esterification, which could lead to tissue lipid accumulation. The authors have to better explain whether and how PPAR-regulation and relative outcome by MA might be health promoting (for example, underlining its connection with inflammation compared to lipd homeostasis) In this connection, plesase nore that lines 392-393 are not clear.
Moreover, as I have already observed, data about MA effectiveness in rescuing tissue from inflammation also are not so consistent, indeed the inflammatory cytokines' profile does not remarkably point for anti-iinflammatory factors' induction as well as pro-inflammatory factors' decrease.
- point 4, the authors have not properly answered. As I had explained in my previous report, oxidative stress evaluation by means of ROS fluorescence determination at the end of treatment is not a good test. For this reason, I think that the relative results cannot support authors' conclusive assessments as regards MA antioxidant properties, at least in their experimental model.
- point 7, "Results" are still lacking comments about DAPI evaluation.
Finally, I still cannot view "Supplementary Materials"
Round 3
Reviewer 2 Report
I appreciate the effort of the authors to adequately amend or clarify most of my previous observations, thus improving the overall quality of the paper.
I confirm also to agree with them for the deletion of ROS production data.
In conclusion, this last version of the manuscript can be accepted for publication.